# Molecular mechanisms of microbiome modulation by the eukaryotic secondary metabolite azelaic acid

**Ahmed A Shibl[1][†], Michael A Ochsenkühn[1], Amin R Mohamed[1], Ashley Isaac[1,2], Lisa SY Coe[1], Yejie Yun[1][‡], Grzegorz Skrzypek[3], Jean-Baptiste Raina[4], Justin R Seymour[4], Ahmed J Afzal[1], Shady A Amin[1,5,6]***

[1]Biology Program, New York University Abu Dhabi, Abu Dhabi, United Arab Emirates; [2]Max Planck Institute for Marine Microbiology, Bremen, Germany; [3]West Australian Biogeochemistry Centre, School of Biological Sciences, The University of Western Australia, Perth, Australia; [4]Climate Change Cluster, Faculty of Science, University of Technology Sydney, Ultimo, Australia; [5]Center for Genomics and Systems Biology (CGSB), New York University Abu Dhabi, Abu Dhabi, United Arab Emirates; [6]Arabian Center for Climate and Environmental Sciences (ACCESS), New York University Abu Dhabi, Abu Dhabi, United Arab Emirates

***For correspondence:**
samin@nyu.edu

**Present address:** [†]Genetic Heritage Group, Biology Program, New York University Abu Dhabi, Abu Dhabi, United Arab Emirates; [‡]Computational Biology Department, Carnegie Mellon University, Pittsburgh, United States

**Competing interest:** The authors declare that no competing interests exist.

**Abstract** Photosynthetic eukaryotes, such as microalgae and plants, foster fundamentally important relationships with their microbiome based on the reciprocal exchange of chemical currencies. Among these, the dicarboxylate metabolite azelaic acid (Aze) appears to play an important, but heterogeneous, role in modulating these microbiomes, as it is used as a carbon source for some heterotrophs but is toxic to others. However, the ability of Aze to promote or inhibit growth, as well as its uptake and assimilation mechanisms into bacterial cells are mostly unknown. Here, we use transcriptomics, transcriptional factor coexpression networks, uptake experiments, and metabolomics to unravel the uptake, catabolism, and toxicity of Aze on two microalgal-associated bacteria, *Phycobacter* and *Alteromonas*, whose growth is promoted or inhibited by Aze, respectively. We identify the first putative Aze transporter in bacteria, a 'C$_4$-TRAP transporter', and show that Aze is assimilated through fatty acid degradation, with further catabolism occurring through the glyoxylate and butanoate metabolism pathways when used as a carbon source. *Phycobacter* took up Aze at an initial uptake rate of $3.8 \times 10^{-9}$ nmol/cell/hr and utilized it as a carbon source in concentrations ranging from 10 µM to 1 mM, suggesting a broad range of acclimation to Aze availability. For growth-impeded bacteria, we infer that Aze inhibits the ribosome and/or protein synthesis and that a suite of efflux pumps is utilized to shuttle Aze outside the cytoplasm. We demonstrate that seawater amended with Aze becomes enriched in bacterial families that can catabolize Aze, which appears to be a different mechanism from that in soil, where modulation by the host plant is required. This study enhances our understanding of carbon cycling in the oceans and how microscale chemical interactions can structure marine microbial populations. In addition, our findings unravel the role of a key chemical currency in the modulation of eukaryote-microbiome interactions across diverse ecosystems.

## eLife assessment

This study presents **valuable** findings on the contrasting responses of two bacteria to the phytoplankton-derived compound azelaic acid. Metabolomics and transcriptomics evidence **convincingly** shows the assimilation pathway in one marine bacterium and a stress response in a

second bacterium. The study provides evidence that azelaic acid can alter marine microbial community structure in mesocosm experiments, though the mechanisms underlying this shift in community structure remain to be explored in future studies.

## Introduction

Eukaryote-microbiome interactions are among the most diverse associations in marine and terrestrial ecosystems. Chemical exchanges between diverse eukaryotes and their microbiomes are hallmarks of these relations, particularly involving secondary metabolites (*Helliwell et al., 2022*; *Shaffer et al., 2022*). Despite the importance of secondary metabolites as signaling molecules that play critical roles in structuring microbial populations and regulating global biogeochemical cycles, the molecular mechanisms by which identified chemicals regulate microbial relationships in the ocean are largely elusive. The $C_9$ dicarboxylic acid, azelaic acid (or azelate, hereafter Aze), is a ubiquitous, yet enigmatic metabolite produced by photosynthetic organisms, such as plants (*Amin et al., 2016*; *Khakimov et al., 2014*) and phytoplankton (*Shibl et al., 2020*). It is postulated to be a product of the peroxidation of galactolipids (*Zoeller et al., 2012*), yet its exact biosynthetic pathway is unknown. Aze plays a crucial role as an infochemical that primes systemic acquired resistance against phytopathogens (*Jung et al., 2009*; *Spoel and Dong, 2012*; *Wittek et al., 2014*) and influences microbial diversity in soil (*Korenblum et al., 2020*). In marine environments, diatoms secrete Aze to modulate bacterial populations by promoting the growth of symbionts while simultaneously inhibiting opportunists (*Shibl et al., 2020*). Despite this undeniable importance, no Aze uptake proteins are known in bacteria and its assimilation into bacterial cells to promote or inhibit growth is poorly understood. A single gene has recently been identified in *Pseudomonas nitroreducens* that acts as a transcriptional regulator of Aze, *azeR* (*Bez et al., 2020*), yet its regulatory targets are unknown. Understanding the molecular mechanisms that enable the transport and assimilation of Aze into bacterial cells to induce positive or negative phenotypes will help to reveal how eukaryotic hosts influence and control their microbiome.

To examine how bacteria transport and catabolize Aze, we used *Phycobacter azelaicus* (*Coe et al., 2023*) (formerly *Phaeobacter* sp. F10, and hereafter *Phycobacter*) as a model to study how Aze promotes bacterial growth and *Alteromonas macleodii* F12 (hereafter *A. macleodii*) as a model to study the inhibitory effects of Aze (*Shibl et al., 2020*). Both bacteria were isolated from the diatom *Asterionellopsis glacialis,* which was shown to produce Aze to modulate its microbiome (*Shibl et al., 2020*). In previous experiments, Aze caused an increase in cell density of *Phycobacter* relative to controls, while temporarily inhibiting the growth of *A. macleodii* (*Shibl et al., 2020*). A combination of transcriptomics, transcriptional coexpression networks (TCNs) of master transcriptional regulators, metabolomics, and uptake experiments were used to elucidate how these bacteria process Aze. Additionally, in situ mesocosm experiments with seawater microbial communities and *Arabidopsis thaliana* were conducted to examine whether Aze influences microbial diversity in natural environments and enriches microbial taxa that carry genes involved in Aze uptake and response. We discovered the first putative Aze transport system in bacteria and map out how each bacterium responds to and processes Aze through complex regulatory networks.

## Results

### Growth and transcriptional response to Aze

*Phycobacter* incubated with 1 mM Aze as the sole carbon source showed a significant increase in growth relative to no-carbon controls or growth with glutamate as the sole carbon source (*Figure 1—figure supplement 1A*). To elucidate the transcriptional responses of *Phycobacter* to growth on Aze, RNA samples were collected at 0.5 and 8 hr after Aze addition to *Phycobacter* to capture the short-term uptake and longer-term catabolism of Aze (*Figure 1—figure supplement 1B*). To infer possible mechanisms of short-term toxicity by *A. macleodii*, samples were collected at 0.5 hr. After 0.5 hr, *Phycobacter* differentially expressed (DE) 273 genes in response to Aze; this number increased to 558 genes after 8 hr, corresponding to ~20% of all CDS in its genome (*Supplementary file 1*). Approximately 78% of DE genes were upregulated in response to Aze (n=652) and most occurred at 8 hr (n=494). Only 70 genes displayed consistent expression profiles at both 0.5 and 8 hr, suggesting the immediate, short-term response to Aze is different from long-term growth on the substrate. In

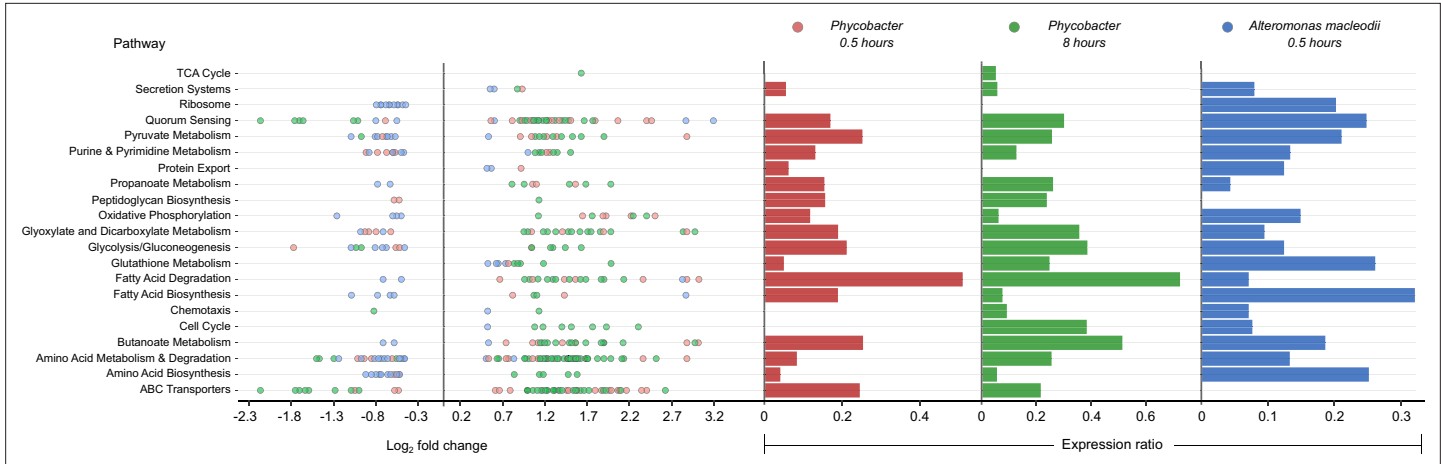

**Figure 1.** Enriched Kyoto Encyclopedia of Genes and Genomes (KEGG) pathways in *Phycobacter* and *A. macleodii*. Left: Scatter plot of the log$_2$ fold-change of differentially expressed (DE) genes present in KEGG pathways listed on the y-axis. Each circle represents the differential expression of a gene in the presence of azelaic acid (Aze) relative to controls. Genes were considered DE if they had a p-adjusted value of <0.05 and log$_2$ fold-change of ≥±0.5. Right: Expression ratios for KEGG pathways of interest were calculated as *the total number of DE genes: total number of genes present in the genome in the given pathway*. Bar sizes are proportional to the contribution of DE genes in relation to all genes present in a pathway. Circle and bar colors indicate the strain and time point (red = *Phycobacter* at 0.5 hr; green = *Phycobacter* at 8 hr; blue = *A. macleodii* at 0.5 hr).

The online version of this article includes the following figure supplement(s) for figure 1:

**Figure supplement 1.** Effect of azelaic acid (Aze) on growth of *Phycobacter*.

contrast, *A. macleodii* DE 274 genes at 0.5 hr, corresponding to ~7% of all CDS in its genome, half of which were upregulated (*Supplementary file 1*).

Global analysis of Kyoto Encyclopedia of Genes and Genomes (KEGG) pathways showed significant differences between the transcriptional profiles of *Phycobacter* and *A. macleodii* (*Figure 1*). While fatty acid degradation, amino acid metabolism and degradation, ABC transporters, oxidative phosphorylation, propanoate metabolism, glyoxylate, and dicarboxylate metabolism pathways exhibited similar expression ratios across both bacteria in response to Aze, upregulated genes in these pathways were mostly dominated by *Phycobacter* transcripts. In contrast, downregulated genes were mostly enriched in *A. macleodii* transcripts. Interestingly, fatty acid degradation genes were among the most upregulated genes in *Phycobacter* (*Figure 1*).

## Aze catabolism by *Phycobacter*

No uptake genes are known for Aze. Among 26 enriched transporter genes in our dataset, a C$_4$-dicarboxylate tripartite ATP-independent periplasmic (TRAP) transporter substrate-binding protein (INS80_RS11065) was the most and the third most upregulated gene in *Phycobacter* grown on Aze at 0.5 and 8 hr, respectively. The small and large permease genes (INS80_RS11060 and INS80_RS11055) neighboring this gene, which form a functional TRAP transporter, were among the top 20 most upregulated genes in the transcriptome at 0.5 hr and continued to be upregulated at 8 hr, implicating these genes in transporting Aze (*Figure 2*, *Supplementary files 2 and 3*). All three genes are co-localized with the putative Aze transcriptional regulator, *azeR* (INS80_RS11050, belonging to the *lclR* family transcriptional factor [TF]) (*Bez et al., 2020*), which was also upregulated at 0.5 hr. In silico predictions revealed that *azeR*, the small and large permeases, all fall within a single operon while the substrate-binding gene is transcribed independently (*Figure 3*). We assign this cluster of genes the designation *azeTSLR* (T=substrate-binding protein, S=small permease, L=large permease, R=regulator). Once inside the cell, Aze appeared to be fed into the fatty acid degradation pathway via a PaaI family thioesterase and acyl-coA ligase (*fadD*) that are present directly downstream of *azeTSLR* and upregulated at 0.5 hr (*Figures 2 and 3*, *Supplementary file 2*). Aze-coA was then putatively degraded through a series of steps catalyzed by genes in the fatty acid degradation pathway (*acd*, *paaF*, and *fadN*) that were among the most highly upregulated genes at 0.5 hr and continued to be upregulated at 8 hr (*Figure 2*, *Supplementary files 2 and 3*). These successive reactions presumably liberated two acetyl-coA molecules to generate glutaryl-coA. This molecule can either be converted

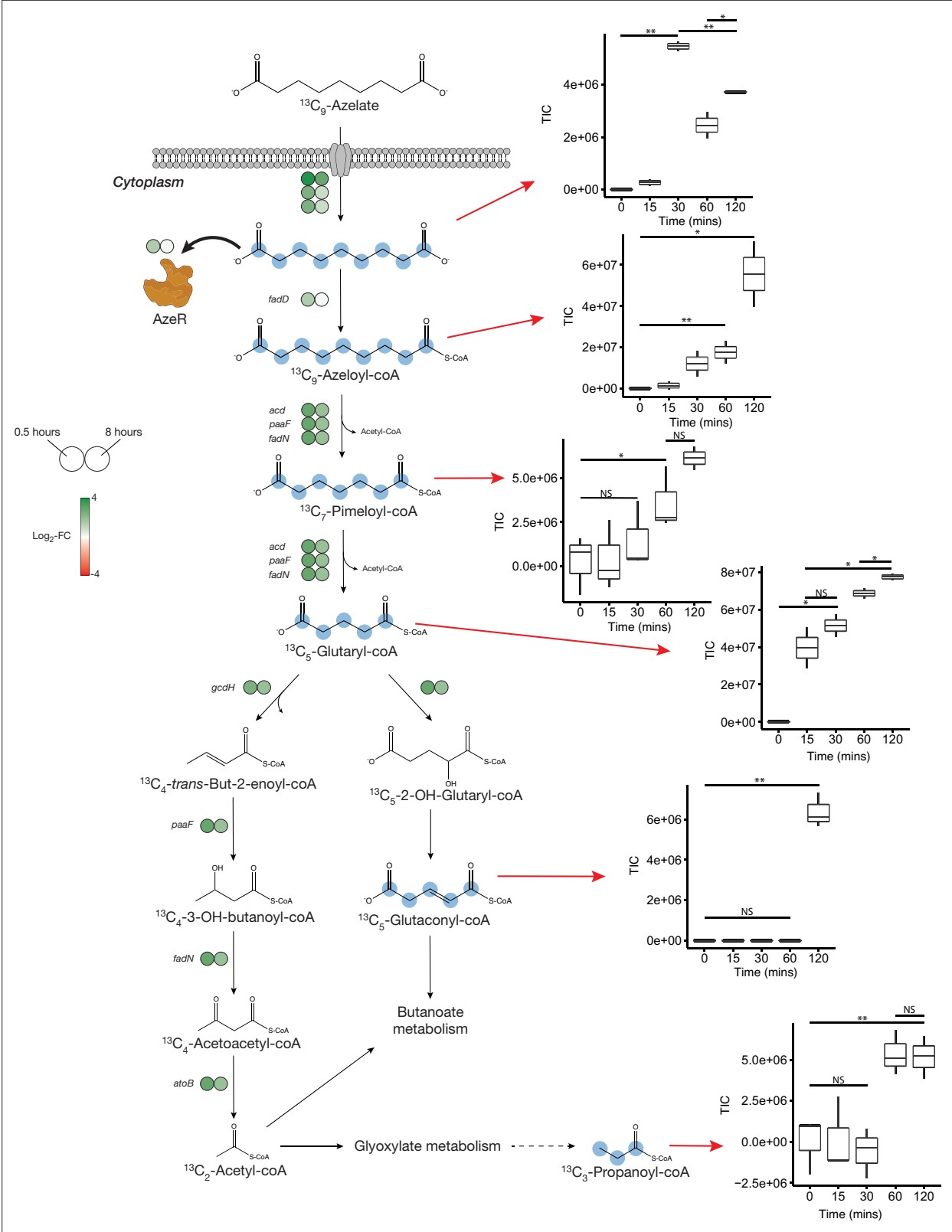

**Figure 2.** Proposed azelaic acid (Aze) catabolism in *Phycobacter* based on transcriptomics and isotope labeling. Left: Log$_2$ fold-change of differentially expressed (DE) genes is shown as circles at 0.5 (left circle) and 8 hr (right circle) after the addition of Aze to *Phycobacter* cells relative to controls. The Aze transport system (*azeTSL*) consists of three genes, the DE values of which are shown next to the transporter. Intermediate reactions and substrates for the successive liberation of acetyl-coA from azeloyl-coA and pimeloyl-coA are not shown; DE values of the genes involved are shown next to each

*Figure 2 continued on next page*

*Figure 2 continued*

overall reaction. $^{13}$C-labeled metabolites detected in the intracellular metabolome of *Phycobacter* cells are marked with cyan circles at each labeled carbon atom site. Right: Relative abundance of each detected labeled metabolite after addition of 100 µM $^{13}$C-Aze to *Phycobacter* cells, shown as total ion current (TIC). Box plot values are based on triplicates. NS indicates no significant relative abundance, * denotes p<0.05 and ** denotes p<0.005 based on a Student's *t*-test.

The online version of this article includes the following figure supplement(s) for figure 2:

**Figure supplement 1.** $^{13}$C-Aze assimilation by *Phycobacter* and *A. macleodii* (reported as δ($^{13}$C) [‰, VPDB]) during a 2 hr incubation.

to glutaconyl-coA, or further degraded into acetyl-coA via *gcdH*, *paaF*, *fadN*, and *atoB*, all of which were upregulated at both time points (**Figure 2**).

To confirm the patterns observed in the *Phycobacter* transcriptome and resolve the fate of glutaryl-coA, *Phycobacter* was supplemented with $^{13}$C$_9$-Aze, and intracellular metabolites were extracted and analyzed using an elemental analyzer-isotope ratio mass spectrometer (IRMS) and a Fourier-transform ion cyclotron resonance mass spectrometer (FT-MS). Within 15 min of $^{13}$C$_9$-Aze addition, *Phycobacter* cells were significantly enriched in $^{13}$C and continued to exhibit an increase in δ($^{13}$C) up to 120 min as shown using IRMS (repeated-measure ANOVA, p<0.001) (**Figure 2—figure supplement 1**). Within 15 min, 0.18% of the carbon content of *Phycobacter* cells was derived from Aze, corresponding to an uptake rate of 3.8×10$^{-9}$ nmol/cell/hr. After 120 min, the fraction of carbon

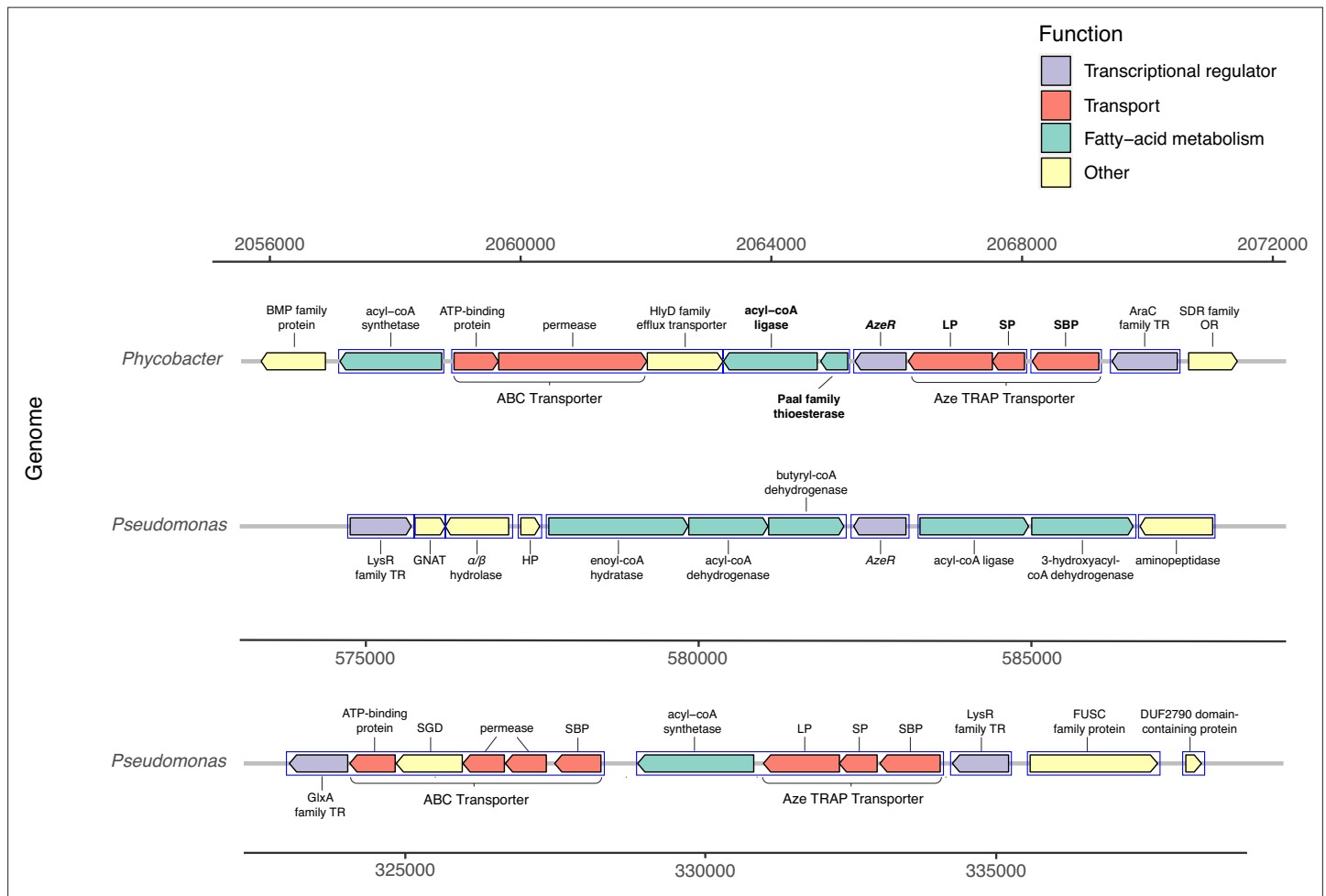

**Figure 3.** Genomic neighborhood structure of *azeR* and *azeTSL* in *Phycobacter* and *P. nitroreducens* DSM9128. Genes predicted in silico to belong to a single operon are marked by blue boxes. Bold-faced gene abbreviations in *Phycobacter* denote upregulated genes in response to azelaic acid (Aze). Operon prediction was carried out on OperonMapper (**Taboada et al., 2018**). azeT = substrate-binding protein, azeS = small permease, azeL = large permease, azeR = transcriptional regulator, TR = transcriptional regulator, OR = oxidoreductase, HP = hypothetical protein, GNAT = GNAT family *N*-acetyltransferase, SGD = succinylglutamate desuccinylase, SBP = substrate-binding protein.

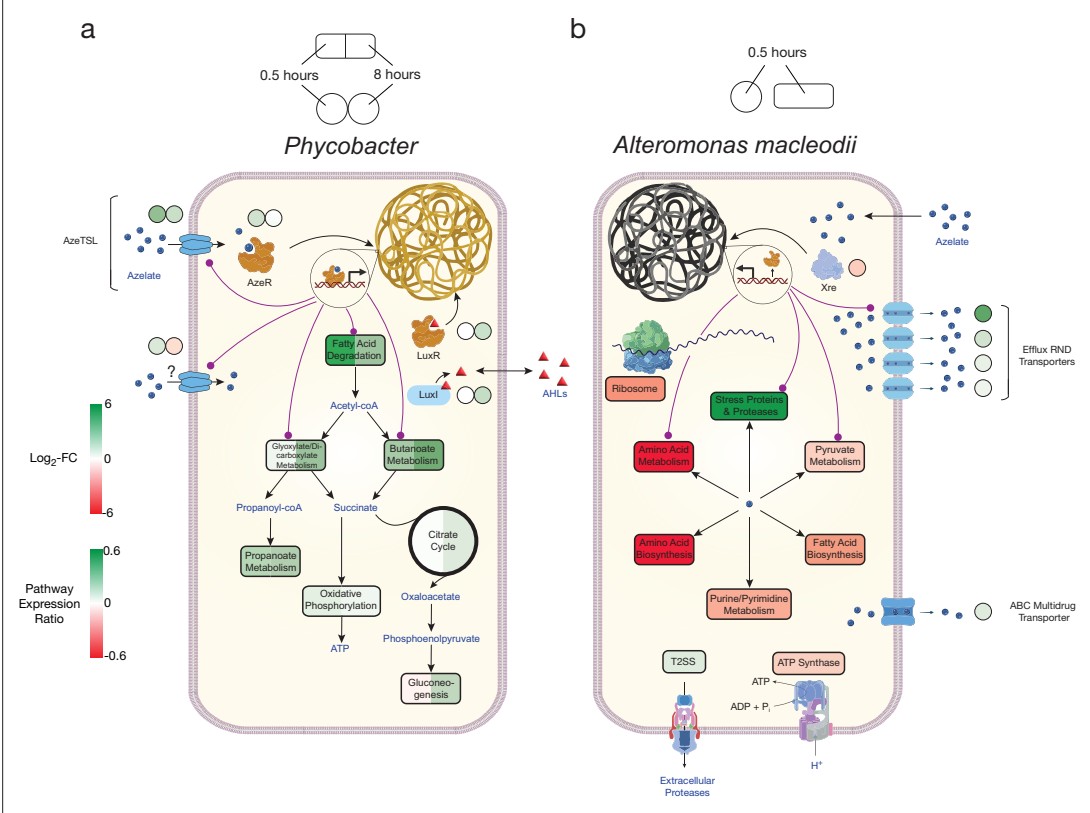

**Figure 4.** Model of azelaic acid (Aze) catabolism in *Phycobacter* and toxic effects in *A. macleodii*. Colored boxes and circles represent metabolic pathway expression ratios and differential gene expression (log$_2$ fold-change), respectively, at 0.5 and 8 hr for (**a**) *Phycobacter* and at 0.5 hr for (**b**) *A. macleodii*. White circles/boxes indicate non-statistically significant differentially expressed (DE) genes/silent pathways. Putative transcriptional factor regulation of pathways/genes is shown by purple lines. Log$_2$ fold-change values shown for all transporters represent the mean log$_2$ fold-change value of all genes in the cluster. Pathway expression ratios were the same as in *Figure 1* except the sign of the ratio (indicating up- or downregulation) are shown. Amino acid biosynthesis and metabolism ratios in (**b**) are outside the range of the pathway expression ratio scale (values ~1.5) but are colored in dark red to indicate significant downregulation. T2SS = Type II secretion system, AHLs = acyl-homoserine lactones.

derived from Aze contributed 0.67% of the total carbon in the cells (*Supplementary file 4*). Consistent with this observation, FT-MS analysis showed that 15–30 min after the addition of labeled Aze to cells, $^{13}C_9$-Aze, $^{13}C_9$-Azeloyl-coA, $^{13}C_7$-pimeloyl-coA, and $^{13}C_5$-glutaryl-coA were detected and increased in relative abundance over the course of 120 min (*Figure 2*, *Supplementary file 4*). Furthermore, $^{13}C_5$-glutaconyl-coA was detected at 2 hr and $^{13}C_3$-propionate-coA at 1–2 hr after addition of $^{13}C_9$-Aze. Glutaconyl-coA is a side product of glutaryl-coA oxidation that is shuttled into butanoate metabolism (*Djurdjevic et al., 2011*), while propionyl-coA is an important intermediate in glyoxylate and propanoate metabolism, indicating these pathways maybe activated downstream of Aze catabolism. Indeed, while greater upregulation of fatty acid degradation occurred at 0.5 hr than at 8 hr, greater upregulation of glyoxylate and dicarboxylate metabolism, butanoate metabolism, and other downstream pathways occurred at 8 hr (*Figure 4a*, *Supplementary file 2*).

## Toxicity of Aze in *A. macleodii*

The genome of *A. macleodii* completely lacks any TRAP transport systems, suggesting Aze crosses the cell membrane non-specifically. In fact, *A. macleodii* assimilated negligible amounts of $^{13}C_9$-Aze, with a relative uptake ~160 times lower than that of *Phycobacter* at 120 min (*Figure 2—figure supplement 1*). In contrast to the *Phycobacter* transcriptome, the highest upregulated gene 0.5 hr after Aze addition to *A. macleodii* was an efflux RND transporter periplasmic subunit (GKZ85_RS10170). The permease and outer membrane subunits of this efflux pump (GKZ85_RS10160, GKZ85_RS10165) that belong to the same operon were among the top 10 most upregulated genes in the *A. macleodii* transcriptome (*Figure 4b*, *Supplementary files 2 and 3*). This efflux system presumably removes Aze from

the cytoplasm. Several other efflux transporters and ABC multidrug transporters were among the most upregulated genes in the presence of Aze relative to the controls. Among upregulated genes in the presence of Aze were secretion systems, protein export, spore formation, stress-response proteins, heat shock proteins, proteases, and ribosome protection genes (*Supplementary files 2 and 3*). Fatty acid degradation was not DE in the presence of Aze, while most ribosomal genes were either down-regulated or not DE. In addition, nucleotide metabolism, pyruvate metabolism, oxidative phosphor-ylation, electron transfer, amino acid metabolism and biosynthesis, and fatty acid biosynthesis were downregulated (*Figures 1 and 4b*, *Supplementary file 2*). Collectively, these transcriptomic patterns confirm previous work (*Shibl et al., 2020*) that indicate *A. macleodii* mitigates the toxic effects of Aze over time and that this mitigation relies on removing Aze from the cell via efflux systems and simul-taneously arresting cellular metabolism. Interestingly, several genes involved in ribosome protection and response to protein degradation (*rmf*, *hpf*, *hslR*, *hslU*, *hslV*) were upregulated, suggesting Aze may disrupt protein synthesis (*Supplementary file 2*).

### *AzeR* and *xre* transcriptional networks

Due to the stark transcriptional response to Aze in each bacterium, we hypothesized that transcrip-tional regulation plays a major role in activating essential pathways that catabolize or mitigate the toxicity of Aze, particularly since both bacteria have complete fatty acid degradation pathways. To identify transcriptional master regulators for *Phycobacter* and *A. macleodii*, we performed regulatory impact factor (RIF) analyses (*Reverter et al., 2010*). TFs in each species were compared to the unique corresponding DE gene lists, which led to the identification of 29 regulators with significant scores for *Phycobacter* and 12 for *A. macleodii* (deviating ±1.96 SD from the mean; p<0.05) (*Supplementary file 5*). Interestingly, *azeR* was the top DE TF identified in *Phycobacter*. In contrast, only two TFs, namely (1) XRE family transcriptional regulator (*xre*), a bacterial TF associated with stress tolerance (*Hu et al., 2018*), and (2) helix-turn-helix transcriptional regulators (GKZ85_RS06065 and GKZ85_RS16495), were DE in *A. macleodii*. To gain insights into the regulatory mechanisms at play during exposure to Aze,

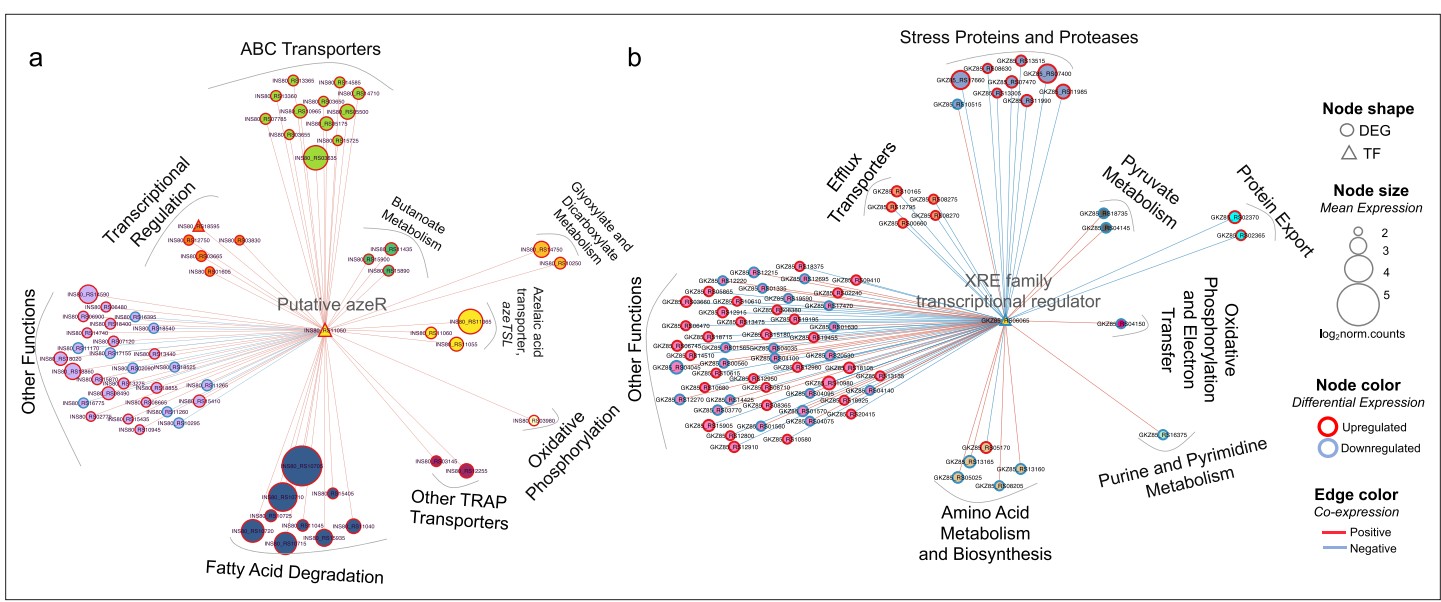

**Figure 5.** Transcriptional coexpression networks of hub transcriptional factors (TFs) *azeR* and *xre*, and their putative target genes in *Phycobacter* and *A. macleodii*. Nodes represent genes (circles) and TFs (triangles) connected by edges based on significant coexpression correlation (PCIT; r≥0.95) for (**a**) *Phycobacter* and (**b**) *A. macleodii*. Nodes are grouped based on functions and represented by different colors. The size of the node corresponds to the normalized mean expression values in azelaic acid (Aze)-treated samples, whereas the color of the node border corresponds to differential expression. Edge colors indicate the direction of the correlation between each gene pair. DEG = differentially expressed gene, TF = transcriptional factor.

The online version of this article includes the following figure supplement(s) for figure 5:

**Figure supplement 1.** Transcriptional coexpression networks constructed using the partial correlation coefficient with information theory algorithm in *Phycobacter* and *A. macleodii*.

we used TCNs, which enabled characterization of transcriptional patterns during response to disease, stress, and development (*Hartl et al., 2021*; *Rose et al., 2016*; *Yao et al., 2015*).

Significant connections (r ≥±0.95) within the initial network identified 286 genes with 7208 connections in *Phycobacter* and 274 genes with 7102 connections in *A. macleodii* (*Figure 5—figure supplement 1*). Subnetworks that contain AzeR and XRE show that they act as hub genes in putatively regulating specific gene categories in *Phycobacter* and *A. macleodii*, respectively. The subnetwork of *azeR* shows that it putatively regulates transcription of most of the pathways that are involved in assimilating Aze in *Phycobacter*, including *azeTSL*, fatty acid degradation, butanoate metabolism, glyoxylate and dicarboxylate metabolism, and oxidative phosphorylation, in addition to other genes that may be indirectly affected by Aze catabolism (*Figures 4a and 5a*). In contrast, the subnetwork of *xre* shows that it putatively regulates transcription of the efflux system implicated in removal of Aze from the cytoplasm of *A. macleodii*, stress proteins and proteases, protein export, nucleotide metabolism, and amino acid metabolism and biosynthesis (*Missiakas et al., 1996*; *Figures 4b and 5b*). These findings highlight the critical role of AzeR in mediating the response to Aze and may explain the large differences in response between both bacteria.

## Aze addition to seawater mesocosms

To examine the influence of Aze as a potential carbon source or antimicrobial compound on marine microbial populations, surface seawater was collected and enriched for prokaryotes by size-fractionated filtration. Seawater mesocosms were supplemented with 100 µM Aze or an equivalent volume of Milli-Q water (see Materials and methods) and incubated in the dark for 16 hr, after which DNA was extracted and 16S ribosomal RNA (rRNA) gene amplicon sequencing was used to assess prokaryotic diversity. The total number of species (richness) was higher in Aze-treated samples relative to T=16 hr control samples (Wilcox, $p<0.05$), while the diversity of the microbial community was significantly lower (Wilcox, $p<0.05$) in Aze-treated samples relative to controls (*Figure 6a*). Principal coordinate analysis (PCoA) showed that the microbial community was significantly different between each sample group (permutational analysis of variance [PERMANOVA], $p<0.001$, *Figure 6b*). Taxonomic classification at the family level showed that Rhodobacteraceae, SAR11 clade I, Cyanobiaceae, and Flavobacteriaceae constituted >60% of prokaryotic diversity across all samples. Notably, the relative abundance of Rhodobacteraceae increased in the Aze-treated samples (mean ~47 %) relative to the T=16 hr control samples (mean ~37 %) (*Figure 6c*). Differential abundance analysis with DESeq2 (p-adj <0.05) identified 44 amplicon sequence variants (ASVs) enriched after Aze addition relative to T=16 hr controls. At the genus level, these ASVs belonged to Rhodobacteraceae (n=17), Marine Group II Euryarchaea (MG II) (n=19), *Pseudoalteromonas* (n=6), *Vibrio* (n=1), and *Psychrosphaera* (n=1). Rhodobacteraceae had the highest mean taxonomic proportion of ASVs ranging from ~1.5 to ~95, suggesting a significant reliance on Aze, while MG II had the lowest mean ratios of ASVs ranging from ~0.02 to ~0.04 (*Figure 6d*).

## Aze addition to soil and infiltration into *Arabidopsis*

We hypothesized that Aze may have similar influences in rhizobial ecosystems as it did in marine ecosystems, specifically it may influence soil microbial community composition. Potting soil was amended with 100 µM Aze, suberic acid, or an equivalent volume of Milli-Q water and incubated in the dark for 16 hr at 24°C. The $C_8$-dicarboxylic acid suberic acid was used to ensure potential changes in the microbial community structure were specific to Aze. DNA was extracted and 16S rRNA gene amplicon sequencing was used to assess prokaryotic diversity. Unlike seawater, Aze-treated soil did not significantly alter the soil microbial community (*Figure 6—figure supplement 1*), which suggests a different mechanism of action. Because Aze is known to prime the immune system in plants, a fundamentally different function from its role in diatoms where it modulates phycosphere microbial communities, we hypothesized that Aze may act indirectly on soil communities through the plant host.

Four-week-old *A. thaliana* leaves were infiltrated with 500 µL of 1 mM Aze, suberic acid, or an equivalent volume of MES buffer as a control and incubated for 5 days. Rhizosphere microbial communities were collected, DNA was extracted, and 16S rRNA gene amplicon sequencing was used to assess prokaryotic diversity. Aze-treated plants displayed lower diversity and richness indices compared to controls (MES buffer and suberic acid) (*Figure 6—figure supplement 2*), although these trends were not statistically significant. Taxonomic classification at the family level revealed a highly diverse

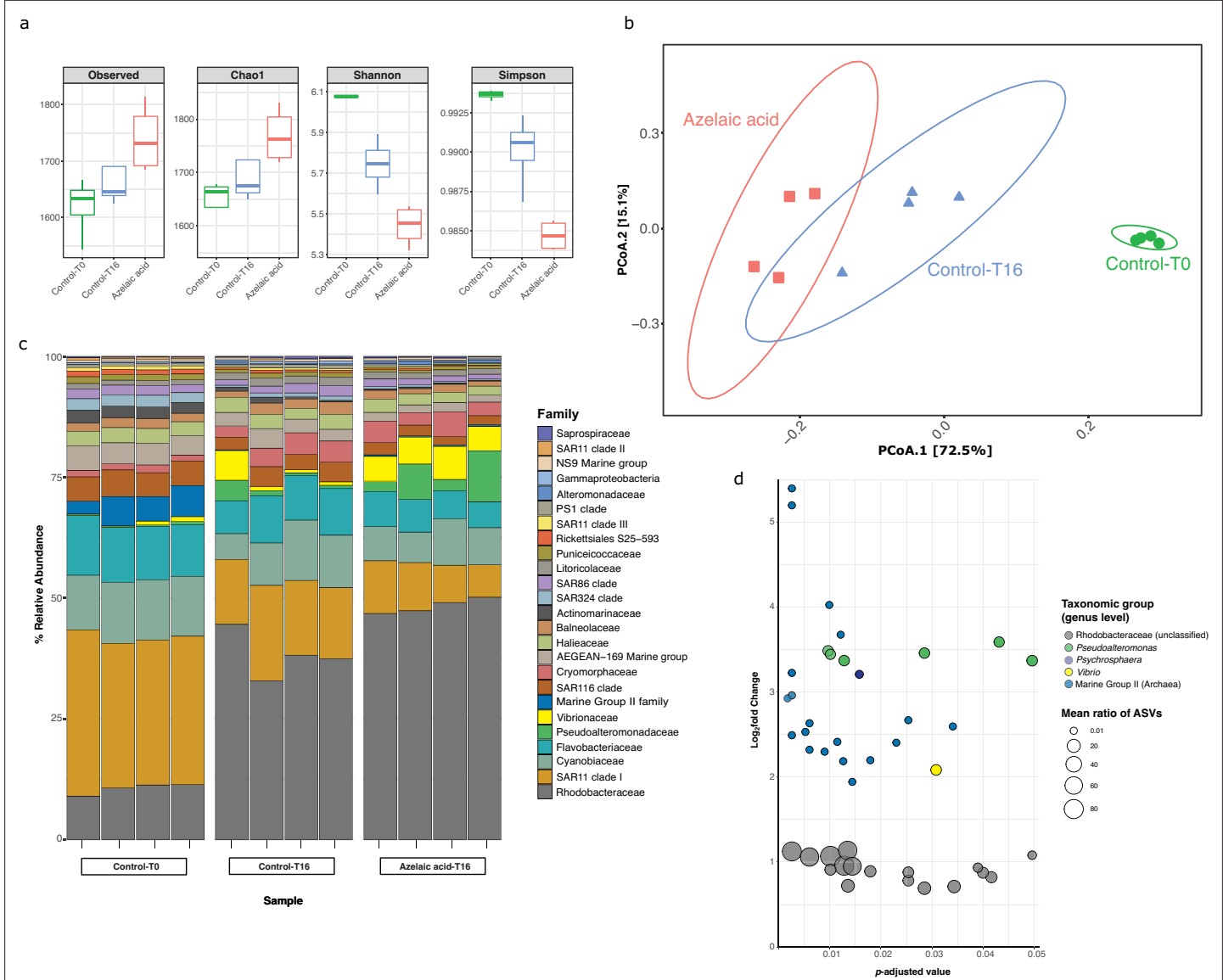

**Figure 6.** Bacterial and archaeal diversity in seawater treated with azelaic acid (Aze). (**a**) Alpha-diversity indices of observed OTUs, Chao1, Shannon and Simpson across the Aze-treated and control samples. (**b**) Principal coordinate analysis (PCoA) of Bray-Curtis distances (permutational analysis of variance [PERMANOVA]: $R^2=0.73$; p<0.001) between samples. The two principal components (PCoA1 and PCoA2) explained 72.5% and 15.1% variance, respectively. (**c**) Relative abundance of the top 25 microbial families based on 16S rRNA gene amplicon sequencing of Aze-treated samples at T=16 hr and control samples at T=0 and T=16 hr. (**d**) Distribution of the amplicon sequence variants (ASVs) belonging to significantly differentially abundant taxa between the Aze-treated and control samples at T=16 hr, according to their log2 fold-change and p-adjusted values. The bubble size indicates the mean taxonomic proportion of each ASV, calculated as *the mean number of reads in an ASV: mean number of reads present in all ASVs of the same taxonomic classification*. The bubble color indicates the taxonomic classification of each ASV according to (**c**).

The online version of this article includes the following figure supplement(s) for figure 6:

**Figure supplement 1.** Effect of azelaic acid (Aze) treatment on the alpha- and beta-diversity of soil.

**Figure supplement 2.** Effect of azelaic acid (Aze) treatment on the alpha- and beta-diversity of *A. thaliana* root microbiome.

**Figure supplement 3.** Bacterial diversity in the roots of *A. thaliana* treated with azelaic acid (Aze).

microbial community across all samples (*Figure 6—figure supplement 3*), but Aze-treated plants showed a distinct root microbiome compared to controls (PERMANOVA, p<0.05; *Figure 6—figure supplement 2*). Differential abundance analysis with DESeq2 (p-adj <0.05) identified 75 and 78 ASVs enriched between Aze vs MES controls and Aze vs suberic acid controls, respectively (*Figure 6—figure supplement 3*). Interestingly, many of the enriched ASVs belonged to typical rhizobial symbionts, such

as Azospirillaceae and Rhizobiaceae. These findings suggest that Aze does not directly influence the soil microbiome but acts indirectly by priming the immune system of *A. thaliana*, which ultimately modifies rhizosphere composition.

## Discussion

Aze is a ubiquitous metabolite produced endogenously by photosynthetic organisms. It was recently revealed that Aze production by a ubiquitous diatom enables it to selectively promote growth of its bacterial symbionts while simultaneously inhibiting growth of opportunists (*Shibl et al., 2020*). In addition, plants use Aze to prime their immune systems in response to phytopathogens (*Jung et al., 2009*). Its toxic effects against the acne-causing bacterium, *Propionibacterium acnes*, popularized its use in skin care products (*Leeming et al., 1986*). Aze has also been shown to have antitumor effects on cancerous cells, and functions as a competitive inhibitor of mitochondrial oxidoreductases and steroid biosynthesis (*Sieber and Hegel, 2014*). The ability of a single metabolite to selectively inhibit or promote different populations of bacteria simultaneously poses important questions about how this metabolite evolved in the eukaryotic chemical repertoire to influence their microbiome and its role in unicellular versus multicellular phototrophs.

Identifying Aze as a putative substrate of a TRAP transporter system is a novel and exciting result. Despite numerous attempts, our efforts to knock out *azeTSL* in *Phycobacter* failed. Information about bacterial transporters and their substrates remain speculative, and much less is known about substrate specificity for bacterial TRAP transporters than others, such as ABC transporters (*Maqbool et al., 2015*). TRAP transporters consist of a substrate-binding domain and two transmembrane segments (large and small permeases), while lacking the canonical nucleotide-binding domain found in ABC transporters (*Rosa et al., 2018*). Recognized substrates for TRAP transporters vary from sugars, mono- and dicarboxylates, organosulfur molecules, heterocyclic carboxylates, and amino acids. Among dicarboxylates, only $C_4$-compounds, such as succinate, fumarate, and malate, have been identified as substrates for TRAP transporters, which explains the prevalent annotation of '$C_4$-*dicarboxylate transporter*' for proteins in bacterial genomes, including *Phycobacter*. Other than $C_6$-adipate that binds with high affinity to a tripartite tricarboxylate transporter (*Rosa et al., 2017*), it is not known how dicarboxylates with >4 carbons are taken up (*Mulligan et al., 2007*). This makes AzeTSL a putative transporter of the longest chain dicarboxylate molecule known to date. The sheer abundance of dicarboxylate transporters in bacterial genomes begs the question of whether they transport a variety of $C_n$-dicarboxylates, such as $C_5$-glutarate, $C_6$-adipate, $C_7$-pimelate, $C_8$-suberate, $C_9$-azelate, and $C_{10}$-sebacate. Some of these metabolites are produced by primary producers and thus potentially serve as growth factors or signals for heterotrophic bacteria (*Moran et al., 2022*). Therefore, further work is needed to expand on the nature of substrates specific to dicarboxylate TRAP transporters.

We have shown that while *Phycobacter* uses fatty acid degradation and subsequently glyoxylate, dicarboxylate, and butanoate metabolism pathways to catabolize Aze, *A. macleodii* attempts to mitigate the toxicity of Aze by cytoplasm efflux and by downregulating its ribosome and protein synthesis pathways (*Figure 4*). However, the fact that *A. macleodii* possesses a complete fatty acid degradation pathway similar to bacteria that display inhibition by Aze is intriguing, as it is unclear why such bacteria cannot catabolize this metabolite. The answer may lie in the fact that *Phycobacter*, and bacteria that catabolize Aze such as *P. nitroreducens*, possess the TF AzeR. Using RIF analysis, we have shown that *azeR* in *Phycobacter* putatively regulates its fatty acid degradation pathway and *azeTSL*, along with other pathways essential for the catabolism of Aze (*Figure 5a*). By activating uptake, fatty acid degradation and other essential genes, AzeR enables bacteria to efficiently metabolize Aze. In contrast, bacteria deficient in AzeR, such as *A. macleodii,* are presumably unable to activate these pathways rapidly and thus fall susceptible to the toxic effects of Aze. Nevertheless, *A. macleodii* likely uses efflux pumps and the activation of a variety of stress-related pathways mediated by the TF XRE (*Figure 5b*) to recover from Aze-dependent toxicity in a relatively short period of time. While Aze exhibits a wide range of inhibitory activity, in *A. macleodii,* ribosome and/or protein synthesis inhibition seem to contribute to the toxicity associated with Aze, evidenced by the strong downregulation of most ribosomal protein coding genes, protein synthesis genes, and proteases. This finding agrees with previous studies that demonstrated the bacteriostatic effects of Aze on protein, DNA and RNA synthesis in *P. acnes* and *Staphylococcus epidermidis* (*Bojar et al., 1988*; *Bojar et al., 1991*).

Aze addition to seawater mesocosms induced a statistically significant increase in the relative abundance of taxa that are capable of assimilating Aze compared to controls. Although the observed patterns may be driven by Aze activity, other mechanisms may be at play that led to changes in the microbial community, such as interspecies competition. Rhodobacteraceae ASVs comprised the most abundant group to be influenced by Aze additions, highlighting their adaptation to phytoplankton exudates (*Landa et al., 2017*; *Teeling et al., 2012*), while MG II archaea ASVs exhibited the largest enrichment relative to controls (*Figure 6d*). While we cannot discount competitive interactions from inducing the significant enrichment of MG II archaea ASVs, the possibility that Aze may be a growth substrate for this important group is intriguing. The MG II archaea (*Candidatus* Poseidoniales) (*Rinke et al., 2019*) remain uncultivated (*Tully, 2019*), hindering our understanding of their metabolic capacity, though isotope probing and metagenomic analysis showed its assimilation of phytoplankton-derived substrates (*Orsi et al., 2016*). Several MG II genera exhibited chemotaxis toward phytoplankton exudates (*Raina et al., 2022*) and displayed significant correlations to chlorophyll *a* in an oceanographic time-series (*Rinke et al., 2019*). Further work is needed to confirm the link, if any, between Euryarchaeota and Aze. In soil, Aze primes the plant immune response (*Jung et al., 2009*), which likely drives the production of a myriad of metabolites that modulate the rhizosphere microbiome. Indeed, introducing Aze directly to soil had negligible effect on the soil microbiome, while Aze introduction to *A. thaliana* appears to cause significant changes to the rhizosphere composition that may reflect complex interactions between the host and rhizosphere microbes. Further work is needed to clarify the mechanisms enabling plants to modulate rhizosphere microbiomes through Aze.

In summary, photosynthetically derived secondary metabolites play a significant role in structuring microalgal and plant microbiomes. In the oceans, they are hypothesized to influence algal blooms, succession of microbial populations, and global biogeochemical cycles. However, the perception and response of these metabolites in microbial heterotrophs remain largely unknown. This study set out to unravel the molecular mechanisms of Aze on bacteria that enables eukaryotic hosts to modulate their associated microbiome. More broadly, we demonstrate that bacterial lineages occupying different ecosystems, such as the phycosphere surrounding microalgal cells, may have gained an ecological advantage over others by evolving mechanisms that assimilate Aze. These findings expand the understanding of bacterial TRAP transporters and their potential substrates, while also highlighting the importance of linking ubiquitous transporter systems to chemical currencies identified in the ocean. While Aze assimilation represents an important mechanism to structure the microbiome of primary producers, it is likely one of many such metabolites that together work in concert to ensure the maintenance of healthy microbial communities, which ultimately influence higher trophic levels and control global biogeochemical cycling.

## Materials and methods
### Growth of bacterial isolates with Aze

To test if *Phycobacter* could use Aze as a sole carbon course, 2 mL of an overnight culture grown in Zobell marine broth (*Raina et al., 2022*) was centrifuged and washed three times with modified mineral media ($K_2HPO_4$ was used instead of $KH_2PO_4$) (*Zech et al., 2009*) lacking a carbon source. Cells were then used to inoculate 5 mL cultures of filter-sterilized mineral media supplemented with 1 mM Aze, 1 mM L-glutamate, or lacking a carbon source. While 1 mM is typically high compared for environmental concentrations, it is on par with amounts of carbon sources added to bacterial culture. Cultures were incubated in the dark with shaking at 26°C. Growth was monitored in 96-well plates and measured using absorbance at 600 nm ($OD_{600}$) with an Epoch 1 Microplate Spectrophotometer (Biotek).

To test the transcriptional response of *Phycobacter* and *A. macleodii* to Aze, cells were grown as previously described (*Shibl et al., 2020*) and inoculated into cultures at a starting density of ~7.5 × $10^5$ cells/mL and $OD_{600}$ 0.2, respectively, in 10% marine broth. This medium was used instead of a no-carbon-source medium to enable the cells to grow in controls and thus shed light on uptake and assimilation mechanisms of Aze while removing the transcriptional responses of cellular growth/division from the transcriptomes. Cultures were supplemented with either 500 μM filter-sterilized Aze (Sigma-Aldrich) or an equal volume of filter-sterilized Milli-Q water. All cultures were shaken in a shaker-incubator at 26°C. Cells were either centrifuged at 13,000 × *g* or were filtered onto 0.22 μm

Sterivex cartridges (Merck Millipore, Burlington, MA, USA) using a peristaltic pump at a flow rate of 40 mL/min. All cells were flash-frozen in liquid nitrogen and stored at –80°C until RNA extraction.

## RNA extraction and sequencing

Cartridges containing *Phycobacter* cells were thawed on ice before extraction, while cell pellets of *A. macleodii* were processed directly from the freezer. Filter membranes were removed with sterile tweezers from cartridges and placed directly in the RLT lysis buffer (QIAGEN). RNA was extracted from cell pellets and filter membranes using the RNeasy Mini Kit (QIAGEN) according to the manufacturer's protocol. RNA samples were treated with DNase I (Thermo Fisher Scientific) to eliminate genomic DNA contamination and sent to NovogeneAIT Genomics (Singapore) for library preparation using the NEBNext Ultra RNA Library Prep Kit and paired-end (2×150 bp) sequencing on the Illumina NovaSeq 6000 (San Diego, CA, USA) platform. All time points were carried out in three biological replicates except for *Phycobacter* at 0.5 hr, which had six replicates. All samples passed RNA QC except for one control replicate from *Phycobacter* at 0.5 hr.

## RNAseq analysis

Raw reads of *Phycobacter* and *A. macleodii* samples were processed using fastp v0.22 (*Chen et al., 2018*) for quality filtering, adapter removal, and trimming. The resulting sequences were then aligned with their respective genomes (WKFH00000000.1 and CP046140.1) using Bowtie2 v2.3.5 (*Langmead and Salzberg, 2012*). Resulting BAM files were processed on SAMtools v1.12 (*Li et al., 2009*) and read counts were generated with featureCounts v2.0.3 (*Liao et al., 2014*). Differential expression analysis between treatment and control samples was done using DESeq2 v3.14 (*Love et al., 2014*) on R v4.1 (R Core Team). Genes were considered DE if they had a p-adjusted value of <0.05 and $\log_2$ fold-change of $\geq\pm0.5$. Pearson coefficient calculations and correlation plots were done using the R gplots package (*Warnes et al., 2016*). DE genes were fed into the KEGG (*Kanehisa and Goto, 2000*) database to infer pathways. Scatter plots of genes in enriched KEGG pathways and bar plots of their expression ratios were generated using the R ggplot2 package (*Wickham, 2016*).

## Isotope labeling and metabolomics

*Phycobacter* cells were grown as described above. Filter-sterilized Aze-$^{13}C_9$ (Sigma-Aldrich) was added to triplicate cultures with an $OD_{600}$ of 0.3 at a final concentration of 500 µM and cultures were shaken in a shaker-incubator at 26°C. Two mL samples were collected at 0, 15, 30, 60, and 120 min after addition of Aze and centrifuged at 13,000 × *g* for 5 min at 4°C, resuspended and washed with 35 g/L NaCl in PBS, then centrifuged and resuspended in ice-cold 100% methanol. Subsequently, samples were sonicated for 2 min on ice, then centrifuged at 4°C for 5 min. Supernatant was collected and dried under nitrogen flow, then the extract was stored at –20°C until analysis using mass spectroscopy.

Samples for direct infusion measurements are prone to salt interference and thus solid-phase extraction with PPL-bond elute columns (Agilent Technologies, USA) was applied. High-resolution mass spectra were acquired on a Bruker solariX XR FT-MS (Bruker Daltonics GmbH, Germany) equipped with a 7 T superconducting magnet and operated in ESI (+) ionization mode with 0.5 bar nebulizer pressure, 4.0 L dry gas and a 220°C capillary temperature. Source optics parameters were 200 V at capillary exit, 220 V deflector plate, 150 V funnel 1, 15 V skimmer 1, 150 Vpp funnel RF amplitude, octopole frequency of 5 MHz, and an RF amplitude of 450 Vpp. Para Cell parameters were set to –20 V transfer exit lens, –10 V analyzer entrance, 0 V side kick, 3 V front and back trap plates, –30 V back trap plate quench, and 24% sweep excitation power. For sample injection, a Triversera Nanomate (Advion BioSciences Inc, USA) with 1 psi gas pressure and ESI (+) voltage of 1.7 kV was used. Three-hundred and twenty scans were accumulated for each sample with an accumulation time of 200 ms and acquired with a time domain of four mega words over a mass range of m/z 75–1200, at an optimal mass range from 200 to 600 m/z. Spectra were internally calibrated using primary metabolites (e.g. amino acids, organic acids) in Data Analysis 5.0 Software (Bruker Daltonics GmbH, Germany). The FT-MS mass spectra were exported to peak lists with a cutoff signal-to-noise ratio of 4. The masses for relevant $^{13}C$ isotope-labeled metabolites were calculated using enviPat: isotope pattern calculator (*Skrzypek and Dunn, 2020*). MSMS was performed at 15 V, 25 V, and 35 V collision voltage and characteristic fragments identified manually.

## Aze uptake

*Phycobacter* and *A. macleodii* cells were grown as described above. Cells were centrifuged and resuspended in triplicate flasks with 10% Zobell marine broth at an $OD_{600}$ of 0.3. Filter-sterilized Aze-$^{13}C_9$ (Sigma-Aldrich) was added to each culture flask at a final concentration of 10 μM and cultures were incubated in a shaker-incubator at 26°C. A lower concentration of Aze-$^{13}C_9$ was used here to allow for an accurate quantification of its uptake. Triplicate samples (20 mL) were collected at 0, 15, 30, 60, and 120 min after addition of Aze and centrifuged at 5,000 × *g* for 15 min. Samples were subsequently fixed with paraformaldehyde (1% final concentration) in artificial seawater for 1 hr at 4°C. Samples were then washed twice with artificial seawater to eliminate any residual paraformaldehyde and cell pellets were dried in an oven at 60°C for 48 hr. A total of 0.50 mg of each dried cell pellet was weighed in a tin capsule (IVA Analysentechnik, Meerbusch, Germany) and loaded onto the autosampler of an elemental analyzer (EA) connected to the Sercon Model 20-22 (Sercon, UK) IRMS. In EA, samples were combusted in the presence of oxygen added to the helium stream at a temperature of 1000°C in a reactor comprised of $Cr_2O_3$ (Sercon, UK) and silvered oxides of cobalt (Sercon, UK). The yield gases were carried through a reduction reactor (650°C, electrolytic copper, Sercon, UK), water traps (granular magnesium perchlorate, Sercon, UK), and a GC column in a stream of helium (grade 99.999% purity; BOC, Australia, 100 mL/min) (*Skrzypek and Paul, 2006*). All stable isotope results are reported as 1000 of δ($^{13}C$) value in permille (‰) after normalization to VPDB scale (*Skrzypek, 2013*) using international standards provided by the International Atomic Energy Agency (IAEA): δ$^{13}$C-USGS40 (–26.39 ‰), USGS41 (37.63 ‰), NBS22 (–30.03 ‰), USGS24 (–16.05 ‰), IAEA603 (2.46 ‰) (*Skrzypek et al., 2010*). The combined standard measurement uncertainty of the δ($^{13}C$) did not exceed 0.10 ‰ (1σ). To calculate the uptake rate of Aze, we quantified: the amount of carbon in bacterial cells at all time points and determined how much was derived from the medium (stable isotope composition of 1.0852 atm%), and how much was derived from $^{13}$C-Aze (stable isotope composition of 99 atm%). The results were converted to $^{13}$C fractions prior to recalculation of the uptakes of Aze using EasyIsoCalculator spreadsheet (*Skrzypek and Dunn, 2020*).

## Seawater mesocosms and 16S rRNA gene amplicon sequencing

Seawater samples were collected in July 2021 from surface waters off the coast of Saadiyat Island (24°38'28.6"N 54°27'09.4"E), United Arab Emirates, kept in the dark, and immediately brought back to the lab. The seawater was filtered through a 1.2 μm filter (Whatman, UK) to remove most phytoplankton and divided into 12 vessels each containing 30 mL to carry out a mesocosm experiment. Control replicates (n=4 at T=0 hr) were immediately filtered onto a 0.2 μm filter (Whatman, UK) to capture microbial cells. Treatment replicates (n=4) with 100 μM Aze and another set of control replicates (n=4) with an equal volume of Milli-Q water added in lieu of Aze were incubated at 24°C for 16 hr in the dark (T=16 hr). We used 100 μM Aze in this case to avoid potential uptake, degradation, and precipitation of this molecule over the long course of the experiment. After incubation, they were immediately filtered onto a 0.2 μm filter (Whatman, UK). Genomic DNA was extracted immediately after filtration from all filters using the DNeasy PowerWater Kit (QIAGEN) and sent to NovogeneAIT Genomics (Singapore) for 16S rRNA gene amplification using the 515F-Y (5'-GTGYCAGCMGCCGC-GGTAA-3') and 926R (5'-CCGYCAATTYMTTTRAGTTT-3') universal primers and paired-end (2×250 bp) sequencing on the Illumina NovaSeq 6000 (San Diego, CA, USA) platform.

Clean raw reads were processed with the rANOMALY R package (*Theil and Rifa, 2021*) using the DADA2 R package (*Callahan et al., 2016*) to generate ASVs. Taxonomic classification was based on the SILVA v138 database. Alpha-diversity was assessed using the richness indices; observed OTUs and Chao1, and diversity indices; Shannon and Simpson. Beta-diversity was assessed and visualized by PCoA based on pairwise Bray-Curtis distance and tested by the PERMANOVA. The differential abundance of taxa across the treatment and control samples was calculated with DESeq2 v3.14 with a p-adjusted value cutoff of <0.05. All plots were generated using the ggplot2 and phylosmith (*Smith, 2019*) R packages and all statistical analyses were performed on R 4.1.2.

## Soil supplementation, *A. thaliana* Aze infusion, incubations, and 16S rRNA gene amplicon sequencing

Two hundred and fifty milligrams of potting soil (Planting Mix No.1; ACE Hardware) was supplemented with 300 μL of 100 μM Aze, suberic acid, or Milli-Q water and incubated at 24°C, in the dark

for 16 hr. Following incubation, the soil samples were flash-frozen in liquid nitrogen and stored at –80°C until DNA extraction.

*A. thaliana* seeds (Col-0) were sowed into potting soil and allowed to germinate under short day conditions (8 hr light/24°C and 16 hr dark/22°C) in a plant growth chamber (Fitoclima 600, Aralab, Rio de Mouro, Portugal). One-week-old healthy seedlings (n=15) were transplanted into round pots containing ~90 g of fresh potting soil, with one seedling per pot. Three additional pots were designated as 'plant-free' that received no seedlings. Pots were covered with cling film for 48 hr to maintain humidity after transplantation. All pots (including plant-free controls) were spatially randomized in the growth chamber and allowed to grow for three additional weeks.

One millimole treatment stocks of Aze (Sigma-Aldrich) and suberic acid (Sigma-Aldrich) were prepared in 5 mM 2-(*N*-morpholino) ethanesulfonic acid (MES) buffer (Sigma-Aldrich) (pH 5.4) and filter-sterilized through 0.2 µm Sterivex filters. Each plant was syringe-infiltrated with 500 µL Aze, suberic acid (equivalent to 2 µmol), or MES buffer (n=5 for each treatment) through the abaxial surface of three leaves. Infiltrated leaves were gently dried with absorbent paper and marked with a felt-tipped pen. Plant-free pots (n=3) were left untreated, and all pots were incubated for 5 days, as described above. Plant roots and plant-free controls were harvested, and the rhizosphere microbial communities were collected as described in *Lebeis et al., 2015*, with modifications. Briefly, above ground plant organs were removed, and loose soil was physically removed until ~1 mm of soil remained on the root surface. Roots were placed in a 25 mL sterile MES buffer in 50 mL centrifuge tubes. Rhizosphere microbial communities were collected by first vortexing the roots in MES buffer for 15 s at maximum speed and subsequently sonicating for 5 min using an Ultrasonic Cleaner (VWR) at power level 3 to liberate tightly adherent and endophytic microbes. The root system was removed aseptically, and the turbid solution was filtered through a 100 µm nylon filter to remove large debris and plant material. The filtrate was centrifuged at 3200 × *g* for 30 min and most of the supernatant discarded. The pellet was resuspended in the remaining supernatant (~200–400 µL) and transferred to sterile 1.5 mL microfuge tubes and centrifuged for 5 min at 10,000 × *g*. The supernatant was removed, and the remaining pellets were flash-frozen in liquid nitrogen and stored at –80°C until DNA extraction. For plant-free pots, approximately 250 mg of soil was collected 2 cm below the surface, transferred to MES buffer, and processed and stored as described above.

Microbial genomic DNA was isolated using the DNeasy PowerSoil Pro Kit (QIAGEN) according to the manufacturer's instructions and samples were sequenced at NovogeneAIT Genomics (Singapore) using 16S rRNA gene amplification using the primers 515F (5′-GTGCCAGCMGCCGCGGTAA-3′) and 806R (5′-GGACTACHVGGGTWTCTAAT-3′) and paired-end (2×250 bp) sequencing on the Illumina NovaSeq 6000 (San Diego, CA, USA) platform. Clean raw reads were processed with the same pipeline as the seawater samples described above.

## Transcriptional master regulator analysis

We used the RIF algorithm (*Reverter et al., 2010*) to detect TFs with high regulatory potential contributing to the observed transcriptional remodeling upon Aze uptake. RIF is designed to identify key regulatory loci contributing to transcriptome divergence between two biological conditions. Predicted TF lists were obtained from the genome annotation (GFF file) of both *Phycobacter* and *A. macleodii* and normalized data (variance stabilized, $\log_2$-transformed counts) of these genes were retrieved. The same normalization strategy was applied to the DE genes. An expression matrix containing normalized expression of the TF genes and DE genes was subjected to RIF analysis per species. RIF analysis identifies regulators that are consistently differentially coexpressed with highly abundant and highly DE genes (RIF1 metric) and regulators that have the most altered ability to act as predictors of the abundance of DE genes (RIF2 metric). TFs were considered significant when the RIF score deviates ±1.96 standard deviation from the mean (*t*-test, p<0.05).

## Transcriptional coexpression networks

The partial correlation and information theory (PCIT) (*Reverter and Chan, 2008*) has been extensively used for gene network analysis (*Alexandre et al., 2021*; *Botwright et al., 2021*). We utilized PCIT to detect significant connections (edges) between pairs of genes (nodes) while considering the influence of a third player (gene). PCIT calculates pairwise correlations between given gene pairs after considering all possible three-way combinations of genes (triads) present within a gene expression matrix.

The algorithm calculates partial correlations after exploring all triads before determining the significance threshold that depends on the average ratio of partial and direct correlation. The set of key regulatory TFs (identified by RIF analysis) and DE genes per species were used for construction of the networks. Normalized data (variance stabilized, $\log_2$-transformed counts) of these genes were used for network construction. The PCIT-inferred networks were visualized using Cytoscape v3.9 (*Shannon et al., 2003*). From these initial networks, we explored a series of subnetworks; first connections between nodes were considered when the partial correlation r was $>\pm0.95$. From these networks, hub genes (potential regulatory components within the network) and their connected genes (first neighbors) were extracted based on: (1) key regulatory factors (with highest RIF scores), (2) differential expression significance, and (3) degree centrality (the number of connections of a node with other nodes in the network).

## Acknowledgements

This work was supported by a grant from the Gordon and Betty Moore Foundation to SAA (GBMF9335, https://doi.org/10.37807/GBMF9335) and by a grant from NYU Abu Dhabi to SAA (AD179). We thank the NYU Abu Dhabi Core Technology Platforms for mass spectrometry support. We also thank Dain McParland for helping collect seawater samples.

## Additional information

### Funding

| Funder | Grant reference number | Author |
| --- | --- | --- |
| Gordon and Betty Moore Foundation | Symbiosis Model Systems GBMF9335 | Shady A Amin Justin R Seymour Jean-Baptiste Raina |
| New York University Abu Dhabi | Faculty Award AD179 | Shady A Amin |

The funders had no role in study design, data collection and interpretation, or the decision to submit the work for publication.

### Author contributions

Ahmed A Shibl, Conceptualization, Data curation, Formal analysis, Supervision, Validation, Investigation, Visualization, Methodology, Writing – original draft; Michael A Ochsenkühn, Formal analysis, Validation, Investigation, Visualization, Methodology, Writing – original draft, Writing – review and editing; Amin R Mohamed, Formal analysis, Investigation, Visualization, Methodology, Writing – original draft, Writing – review and editing; Ashley Isaac, Formal analysis, Visualization, Methodology, Writing – original draft, Writing – review and editing; Lisa SY Coe, Data curation, Methodology, Writing – original draft, Writing – review and editing; Yejie Yun, Jean-Baptiste Raina, Formal analysis, Visualization, Methodology, Writing – review and editing; Grzegorz Skrzypek, Formal analysis, Investigation, Methodology, Writing – review and editing; Justin R Seymour, Funding acquisition, Writing – review and editing; Ahmed J Afzal, Conceptualization, Investigation, Project administration, Writing – review and editing; Shady A Amin, Conceptualization, Resources, Formal analysis, Supervision, Funding acquisition, Visualization, Writing – original draft, Project administration, Writing – review and editing

### Author ORCIDs

Ahmed A Shibl ⓘ http://orcid.org/0000-0002-8147-8406
Michael A Ochsenkühn ⓘ http://orcid.org/0000-0003-0878-1756
Ashley Isaac ⓘ http://orcid.org/0000-0001-9859-4190
Grzegorz Skrzypek ⓘ http://orcid.org/0000-0002-5686-2393
Jean-Baptiste Raina ⓘ http://orcid.org/0000-0002-7508-0004
Shady A Amin ⓘ http://orcid.org/0000-0003-3780-8102

Reviewer #1 (Public Review): https://doi.org/10.7554/eLife.88525.3.sa1

Reviewer #2 (Public Review): https://doi.org/10.7554/eLife.88525.3.sa2
Author Response https://doi.org/10.7554/eLife.88525.3.sa3

## Additional files

### Supplementary files
• Supplementary file 1. A summary of the differentially expressed (DE) genes in *Phycobacter* and *A. macleodii*. The first worksheet summarizes the number of DE genes for *Phycobacter* at 0.5 and 8 hr and *A. macleodii* at 8 hr in response to azelaic acid. Percent values represent the ratio of genes in each category to the total number of genes in each genome. The second, third, and fourth sheets provide the DESeq output data (product ID, baseMean, log2FoldChange, pvalues, etc.) for *Phycobacter* at 0.5 and 8 hr and *A. macleodii* at 8 hr.

• Supplementary file 2. List of selected differentially expressed (DE) genes by (A) *Phycobacter* and (B) *A. macleodii* in response to azelaic acid addition. Genes with no values indicate no statistically significant differential expression compared to controls. Genes were considered significantly DE if they had a p-adjusted value of <0.05 and a $\log_2$ fold-change of $\geq\pm0.50$. Genes predicted to be in a single operon structure are listed in the direction of transcription within a single box, where relevant. Genes operon structures were predicted by OperonMapper. Genes common to more than one pathway are listed only once.

• Supplementary file 3. List of all differentially expressed (DE) genes by (A) *Phycobacter* and (B) *A. macleodii* in response to azelaic acid addition. Genes were considered significantly DE if they had a p-adjusted value of <0.05 and a $\log_2$ fold-change of $\geq\pm0.50$.

• Supplementary file 4. Characteristics of $^{13}$C-Aze uptake by *Phycobacter* and *A. macleodii*. (A) Carbon fractionation and uptake rate of bacteria exposed to $^{13}$C-Aze. (B) $^{13}$C-labeled metabolites detected in *Phycobacter* after incubation with $^{13}$C-Aze.

• Supplementary file 5. Transcriptional factors (TFs) with significant regulatory impact factor (RIF) score and their differential gene expression in *Phycobacter* and *A. macleodii*.

• MDAR checklist

### Data availability
RNAseq raw reads of Phycobacter are deposited in NCBI under the BioProject number PRJNA823575. RNAseq raw reads of A. macleodii are deposited in NCBI under the BioProject number PRJNA823732. 16S rRNA gene amplicon sequencing raw reads from the seawater mesocosm and Arabidopsis root microbiome are deposited in NCBI under the BioProject numbers PRJNA823745 and PRJNA841046, respectively. Mass spectral datasets are available in the MassIVE database under accession MSV000089221. All software packages used in this study are open source. Bacterial strains used are available upon request from the corresponding author.

The following datasets were generated:

| Author(s) | Year | Dataset title | Dataset URL | Database and Identifier |
|---|---|---|---|---|
| Shibl A, Amin SA | 2022 | Phaeobacter F10 azelaic acid RNA-seq | https://www.ncbi.nlm.nih.gov/bioproject/PRJNA823575/ | NCBI BioProject, PRJNA823575 |
| Shibl A, Amin SA | 2022 | Alteromonas macleodii F12 azelaic acid RNA-seq experiment | https://www.ncbi.nlm.nih.gov/bioproject/?term=PRJNA823732 | NCBI BioProject, PRJNA823732 |
| Shibl A, Amin SA | 2022 | 16S Amplicon Metagenomic Sequencing | https://www.ncbi.nlm.nih.gov/bioproject/?term=PRJNA823745 | NCBI BioProject, PRJNA823745 |
| Isaac A, Amin SA | 2022 | Effect of Azelaic acid on plant root microbiome | https://www.ncbi.nlm.nih.gov/bioproject/?term=PRJNA841046 | NCBI BioProject, PRJNA841046 |

*Continued on next page*

*Continued*

| Author(s) | Year | Dataset title | Dataset URL | Database and Identifier |
|---|---|---|---|---|
| Ochsenkühn MA, Amin SA | 2022 | Select bacterial families assimilate the primary producer metabolite, azelaic acid | https://massive.ucsd.edu/ProteoSAFe/dataset.jsp?task=4fa71b2941bf424f8eb2c82e807ecc43 | MassIVE, MSV000089221 |

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
