## [Editor Report · eLife assessment]

This study presents **valuable** findings on the contrasting responses of two bacteria to the phytoplankton-derived compound azelaic acid. Metabolomics and transcriptomics evidence **convincingly** shows the assimilation pathway in one marine bacterium and a stress response in a second bacterium. The study provides evidence that azelaic acid can alter marine microbial community structure in mesocosm experiments, though the mechanisms underlying this shift in community structure remain to be explored in future studies.

---

## [Referee Report · Reviewer #1 (Public Review)]

Summary:

Shibl et al., studied the possible role of dicarboxylate metabolite azelaic acid (Aze) in modulating the response of different bacteria, it was used as a carbon source by Phycobacter and possibly toxic for Alteromonas. The experiments were well conducted using transcriptomics, transcriptional factor coexpression networks, uptake experiments, and chemical methods to unravel the uptake, catabolism, and toxicity of Aze on these two bacteria. They identified a putative Aze TRAP transporter in bacteria and showed that Aze is assimilated through fatty acid degradation in Phycobacter. Meanwhile, in Alteromonas it is suggested that Aze inhibits the ribosome and/or protein synthesis, and that efflux pumps shuttles Aze outside the cytoplasm. Further on, they demonstrate that seawater amended with Aze selects for microbes that can catabolize Aze.

Major strengths:

The manuscript is well written and very clear. Through the combination of gene expression, transcriptional factor co-expression networks, uptake experiments, and chemical methods Shibl et al., showed that Aze has a different response in two bacteria.

Major weakness:

There is no phenotype confirmation of the Aze TRAP transporters through mutagenesis.

Impact on the field:

Metabolites exert a significant influence on microbial communities in the ocean, playing a crucial role in their composition, dynamics, and biogeochemical cycles. This research highlights the intriguing capacity of a single metabolite to induce contrasting responses in distinct bacterial species, underscoring its role in shaping microbial interactions and ecosystem functions.

---

## [Referee Report · Reviewer #2 (Public Review)]

This study explores the breadth of effects of one important metabolite, azelaic acid, on marine microbes, and reveals in-depth its pathway of uptake and catabolism in one model bacterial strain. This compound is known to be widely produced by phytoplankton and plants, and to have complex effects on associated microbiomes.

This work uses transcriptomics to assay the response of two strains that show contrasting responses to the metabolite: one catabolizes the compound and assimilates the carbon, while the other shows growth inhibition and stress response. A highly induced TRAP transporter, adjacent to a previously identified regulator, is inferred to be the specific uptake system for azelaic acid, though this function was not directly tested via genetic or biochemical methods. Nevertheless, this is a significant finding that will be useful for exploring the distribution of azelaic acid uptake capability across metagenomes and other bacteria.

The authors use pulse-chase style metabolomics experiments to beautifully demonstrate the fate of azelaic acid through catabolic pathways. They also measure an assimilation rate per cell, though it remains unclear how this measured rate relates to natural systems. The metabolomics approach is an elegant way to show carbon flux through cells, and could serve as a model for future studies.

The study seeks to extend the results from two model strains to complex communities, using seawater mesocosm experiments and soil/Arabidopsis experiments. The seawater experiments show a community shift in mesocosms with added azelaic acid. The mechanisms for the shift were not determined in this study; further work is necessary to demonstrate which community members are directly assimilating the compound, benefitting indirectly, or experiencing inhibition. The authors also took the unusual and creative step of performing similar experiments in a soil - Arabidopsis system. I admire the authors' desire to identify unifying themes across ecosystems. The parallels are intriguing, and future experiments could determine the different modes of action in aquatic vs terrestrial microbial communities.

This work is a nice illustration of how we can begin to tease apart the effects of chemical currencies on marine ecosystems. A key strength of this work is the combination of transcriptomics and metabolomics methods, along with assaying the impacts of the metabolite on model strains of bacteria and whole communities. Given the sheer number of compounds that probably play critical roles in community interactions, a key challenge for the field will be navigating the tradeoffs between breadth and depth in future studies of metabolite impacts. This study offers a good compromise and will be a useful model for future studies.

---

## [Author Response]

The following is the authors’ response to the original reviews.

**Public Reviews:**

**Reviewer #1 (Public Review):**
Summary:Shibl et al., studied the possible role of dicarboxylate metabolite azelaic acid (Aze) in modulating the response of different bacteria, it was used as a carbon source by Phycobacter and possibly toxic for Alteromonas. The experiments were well conducted using transcriptomics, transcriptional factor coexpression networks, uptake experiments, and chemical methods to unravel the uptake, catabolism, and toxicity of Aze on these two bacteria. They identified a putative Aze TRAP transporter in bacteria and showed that Aze is assimilated through fatty acid degradation in Phycobacter. Meanwhile, in Alteromonas it is suggested that Aze inhibits the ribosome and/or protein synthesis, and that efflux pumps shuttles Aze outside the cytoplasm. Further on, they demonstrate that seawater amended with Aze selects for microbes that can catabolize Aze.Major strengths:The manuscript is well written and very clear. Through the combination of gene expression, transcriptional factor co-expression networks, uptake experiments, and chemical methods Shibl et al., showed that Aze has a different response in two bacteria.Major weakness:There is no confirmation of the Aze TRAP transporters through mutagenesis.Impact on the field:Metabolites exert a significant influence on microbial communities in the ocean, playing a crucial role in their composition, dynamics, and biogeochemical cycles. This research highlights the intriguing capacity of a single metabolite to induce contrasting responses in distinct bacterial species, underscoring its role in shaping microbial interactions and ecosystem functions.

We thank the reviewer for their comments on the paper and we appreciate their suggestion to confirm the activity of Aze TRAP transporters through mutagenesis. We agree that this would be a valuable addition to the study, and we mention in the text that “Despite numerous attempts, our efforts to knock-out azeTSL in Phycobacter failed.”

The success rate of mutagenesis experiments is often low and time-consuming. There are a few reasons why our knock-out experiments with Phycobacter have not been successful. Despite using several modified protocols for electroporation, no Phycobacter colonies grew on the antibiotic plate. We then tried the homologous recombination approach for conjugation but were not successful in selecting for Phycobacter cells, even when grown in high salinity conditions that favor Phycobacter and disfavor the carrier, *E. coli* . While we would love to include a mutagen to confirm the function of this cluster, the task seems to be unattainable at the moment.

**Reviewer #2 (Public Review):**
This study explores the breadth of effects of one important metabolite, azelaic acid, on marine microbes, and reveals in-depth its pathway of uptake and catabolism in one model bacterial strain. This compound is known to be widely produced by phytoplankton and plants, and to have complex effects on associated microbiomes.This work uses transcriptomics to assay the response of two strains that show contrasting responses to the metabolite: one catabolizes the compound and assimilates the carbon, while the other shows growth inhibition and stress response. A highly induced TRAP transporter, adjacent to a previously identified regulator, is inferred to be the specific uptake system for azelaic acid. However the transport function was not directly tested via genetic or biochemical methods. Nevertheless, this is a significant finding that will be useful for exploring the distribution of azelaic acid uptake capability across metagenomes and other bacteria.The authors use pulse-chase style metabolomics experiments to beautifully demonstrate the fate of azelaic acid through catabolic pathways. They also measure an assimilation rate per cell, though it remains unclear how this measured rate relates to natural systems. The metabolomics approach is an elegant way to show carbon flux through cells, and could serve as a model for future studies.The study seeks to extend the results from two model strains to complex communities, using seawater mesocosm experiments and soil/Arabidopsis experiments. The seawater experiments show a community shift in mesocosms with added azelaic acid. However, the mechanisms for the shift were not determined; further work is necessary to demonstrate which community members are directly assimilating the compound vs. benefitting indirectly or experiencing inhibition. In my opinion the soil and Arabidopsis experiments are quite preliminary. I appreciate the authors' desire to broaden the scope beyond marine systems, but I believe any conclusions regarding different modes of action in aquatic vs terrestrial microbial communities are speculative at this stage.This work is a nice illustration of how we can begin to tease apart the effects of chemical currencies on marine ecosystems. A key strength of this work is the combination of transcriptomics and metabolomics methods, along with assaying the impacts of the metabolite on both model strains of bacteria and whole communities. Given the sheer number of compounds that probably play critical roles in community interactions, a key challenge for the field will be navigating the tradeoffs between breadth and depth in future studies of metabolite impacts. This study offers a good compromise and will be a useful model for future studies.

We thank the reviewer for their thoughtful comments on the manuscript. We appreciate their feedback on the breadth of effects of Aze on marine microbes, and their insights into the strengths and limitations of our study.

We agree that the specific mechanisms underlying community-level shifts in seawater mesocosm experiments with added Aze are not yet fully understood and we believe such work is beyond the scope of this paper and warrants an in-depth study of its own. This can perhaps be conducted at a larger scale by using a combination of meta-omics and targeted enrichment to identify the community members directly assimilating Aze, as well as those that are benefitting indirectly or experiencing inhibition.

We also agree that the soil and Arabidopsis experiments are exploratory. However, we believe that these experiments are a valuable first step in highlighting the potential for Aze to have different modes of action in aquatic versus terrestrial microbial communities. Our interest in contrasting bacterial molecular responses in terrestrial plant rhizospheres and marine algal phycospheres stems from the fact that both environments share similar molecules and related bacteria, yet exhibit significantly different evolutionary histories and fluid dynamic profiles (Seymour et al 2017, Nature Microbiol ). Although more is known about Aze in Arabidopsis than phytoplankton, there are still gaps in this knowledge. For example, recent work has shown that Aze and derivatives can be secreted into soil (Korenblum et al 2020, PNAS ), but whether Aze directly influences microbial communities in soil as we have shown in seawater has not been explored. Thus, we feel our preliminary experiments in soil are important to provide such a distinction with seawater. Additional studies in these systems to further investigate the importance of Aze, which were beyond the scope of this current work, would be quite beneficial.

**Reviewer #1 (Recommendations For The Authors):**
General comments:A complete supplemental file of differentially expressed genes should be provided in the supplemental. Please add tables with the entire DESeq output for Aze additions in the genomes of Phycobacter (0.5 and 8 h) and Alteromonas (0.5 h). While it makes sense to focus the paper on Aze related genes, the full dataset should be made available in a more curated form than just the raw reads in the SRA.

We thank the reviewer for this suggestion. We have included three more sheets in Supplementary Table 1 file where readers can find the entire DESeq outputs of Phycobacter (0.5 and 8 h) and Alteromonas (0.5 h) experiments.

Specific comments:• L82 indicates the TRAP transporter for Aze. Looking at the table for gene expression ofPhycobacter there are 26 significantly enriched transport genes at 0.5 h other than the putative Aze TRAP transporter. Even though the TRAP transporter is likely transporting Aze, it would be good to let the readers know that other transporters showed transcript enrichment.

Thank you for this helpful comment. We modified the sentence accordingly to read as follows: “Among 26 enriched transporter genes in our dataset, a C 4 - dicarboxylate tripartite ATP-independent periplasmic (TRAP) transporter substrate-binding protein (INS80_RS11065) was the most and the third most upregulated gene in Phycobacter grown on Aze at 0.5 and 8 hours, respectively.”

• Figure 1: There are many genes enriched from -1 to 1. Is there a cut off, p-val (can you add it to the caption)? It would be good to have a dashed line or something that indicates the -1 and 1 log2 fold change in the figure.

We thank the reviewer for this suggestion. We added the following sentence to the legend of Fig. 1: “Genes were considered DE with a p -adjusted value of < 0.05 and a log2 fold-change of ≥ ±0.50.”

• Supplementary tables: Add a title on all the supplementary tables. It's hard to tell what each one of the tables means without looking at the text and content of each tables.

A short descriptive title is now added to all supplementary tables.

• Not sure if it matters, though Table S1 was not available in the attached files, though it is in the complete pdf.

Table S1 is now in the attached files and the DESeq output has been added to it as suggested in the general comment above.

**Reviewer #2 (Recommendations For The Authors):**
Here I offer some more specific suggestions and comments on the methods and presentation.I recommend being careful throughout with the language regarding conclusions. For instance, the study does not directly demonstrate the activity of the TRAP transporter (as mentioned above), and does not directly demonstrate that the bacteria that increase in abundance in the mesocosm experiments are actually assimilating azelaic acid.

We thank the reviewer for this comment. We agree that further studies are required to get definitive answers regarding the direct activity of the transporter genes and direct assimilation of Aze by bacteria in the mesocosm. These complex experiments would require establishing a reproducible workflow for knocking out genes and further isotope labeling experiments to track Aze assimilation in a natural setting. To that end, we were keen on using language throughout the manuscript indicating that transporter activity is putative. We went through the manuscript again to make sure it was clear that the transporter activity is putative at this time and is not confirmed. For the mesocosms, we cannot rule out that the changes in community structure is not due to other factors besides Aze. We have added this sentence in the discussion of the mesocosm experiments to indicate that the observed changes in microbial community cannot be directly attributed to Aze activity and may be a byproduct of other mechanisms.

Additionally, I find the soil and plant experiments to be very preliminary, and would personally recommend removing them from the manuscript. This is of course the authors' choice, but I find they detract from an otherwise more solid story. I wonder whether 16 hours was sufficient to see community changes and whether adding azelaic acid directly into the plant is necessary or relevant. The study does not measure any plant immune responses so I caution against drawing conclusions about the mechanism. It seems the connection to plant immunity was already shown in the literature, in which case I'm not sure whether these experiments presented here really add anything new to the paper.

We thank the reviewer for these comments. Our 16-hour sampling time point (similar to the seawater experiment) represents an overnight incubation period that should allow sufficient change in the natural microbial composition yet avoids the long-term succession of microbes with high metabolic capacities that may outcompete the rest of the community at long incubation periods. Deciding on this length of incubation was also informed by the uptake rate of Aze and its influence on either bacteria assimilating it as a carbon source or being inhibited by it.

Since no significant changes were observed in the soil, it was necessary to test the hypothesis that the plant host might be indirectly influencing the rhizosphere microbial communities by infiltrating *A. thaliana* leaves with Aze. As the reviewer mentions, the association between Aze and plant immunity was previously shown; however, the overall influence on the microbial community has not been fully explored yet. The soil and plant experiments were meant to serve an exploratory purpose and we find them necessary to keep in the manuscript as a first step in comparing the mode of action of Aze within marine and terrestrial ecosystems. They are by no means the answer to what role Aze plays in soil systems, but rather they are the starting point. We hope that our results encourage some readers to investigate similar common metabolites to further elucidate the molecular underpinnings of microbial modulation in both environments.

Regarding the transcriptomics data, I am not clear on why the "expression ratio" -- i.e. the fraction of pathway genes that were differentially abundant -- was used. I would not expect all transcripts in a pathway to behave the same way in response to a perturbation, due to variation in half-life/stability, post-transcriptional and post-translational regulation, etc. I recommend removing the expression ratio (right panel) from Figure 1. The left panel shows the data more clearly and more directly.

We thank the reviewer for their insight and we agree that not all transcripts in a pathway behave the same way. However, we find the expression ratio panel visually informative to highlight the importance of a pathway in response to Aze, taking into consideration the total number of key genes involved in a pathway. For example, despite the larger number of DE genes associated with the Amino Acid Metabolism & Degradation pathway compared to the Fatty Acid Degradation pathway, the expression ratio for the former in each transcriptome is lower than its Fatty Acid Degradation counterpart, indicating that the response of key fatty acid degradation genes to Aze is more pronounced. We have qualified the reasons for including expression ratios in Figure 1 legend.

Overall I enjoyed reading the manuscript and applaud the authors on a nice contribution to this important field.